Gorgonopsian therapsids (Nochnitsa gen. nov. and Viatkogorgon) from the Permian Kotelnich locality of Russia

Kammerer Christian F. 1 christian.kammerer@naturalsciences.org
Masyutin Vladimir 2
1 North Carolina Museum of Natural Sciences , Raleigh, NC , USA
2 Vyatka Paleontological Museum , Kirov , Russia
Young Mark
Electronic publication date: 2018 Jun 8
Publication date: 2018
Volume: 6
Electronic Location ID: e4954
Received 2018 Mar 13; Accepted 2018 May 22
Copyright: © 2018 Kammerer et al.
Copyright year: 2018
Copyright holder: Kammerer et al.
License: This is an open access article distributed under the terms of the Creative Commons Attribution License, which permits unrestricted use, distribution, reproduction and adaptation in any medium and for any purpose provided that it is properly attributed. For attribution, the original author(s), title, publication source (PeerJ) and either DOI or URL of the article must be cited.
License URL: https://creativecommons.org/licenses/by/4.0/

Keywords: Synapsida, Therapsida, Gorgonopsia, Permian, Russia, Phylogeny, Biogeography

Funding: Deutsche Forschungsgemeinschaft KA 4133/1-1 SYNTHESYS Project This work was supported by a grant from the Deutsche Forschungsgemeinschaft (KA 4133/1-1 to Christian F. Kammerer) and support from the SYNTHESYS Project (http://www.synthesys.info) which is financed by European Community Research Infrastructure Action under the FP7 Integrating Activities Programme. The funders had no role in study design, data collection and analysis, decision to publish, or preparation of the manuscript.

==============================
The early evolution of gorgonopsians is poorly understood. New material from the Kotelnich locality in Russia expands our knowledge of middle/earliest late Permian gorgonopsians from Laurasia. Two gorgonopsian taxa are recognized from Kotelnich: Viatkogorgon ivakhnenkoi Tatarinov, 1999 and Nochnitsa geminidens gen. et sp. nov. Nochnitsa can be distinguished from all known gorgonopsians by its unique upper postcanine tooth row, composed of pairs of teeth (a small anterior and larger posterior) separated by diastemata. Both Viatkogorgon and Nochnitsa are relatively small gorgonopsians, comparable in size to the South African middle Permian taxon Eriphostoma. Inclusion of Viatkogorgon and Nochnitsa in a phylogenetic analysis of gorgonopsians recovers them in basal positions, with Nochnitsa representing the earliest-diverging gorgonopsian genus. All other sampled gorgonopsians fall into two major subclades: one made up entirely of Russian taxa (Inostrancevia, Pravoslavlevia, Sauroctonus, and Suchogorgon) and the other containing only African gorgonopsians. The high degree of endemism indicated in this analysis for gorgonopsians is remarkable, especially given the extensive intercontinental dispersal inferred for coeval therapsid groups.

Introduction

The therapsid subclade Gorgonopsia was an abundant but morphologically conservative group of saber-toothed carnivores that included the apex predators of the late Permian (Sigogneau-Russell, 1989; Kammerer, 2015, 2016). Despite their omnipresence in the late Permian therapsid faunas of southern Africa (Sigogneau, 1970; Smith, Rubidge & van der Walt, 2012), gorgonopsians are poorly represented in the global fossil record. No gorgonopsian fossils have thus far been found in the middle-to-late Permian deposits of western Europe, South America, or southeast Asia (Benton & Walker, 1985; Sues & Munk, 1996; Bercovici et al., 2012; Dias-da-Silva, 2012; Boos et al., 2015). Young (1979) described a supposed gorgonopsian taxon (Wangwusaurus tayuensis) from the Jiyuan Formation of China, but the type material of this taxon consists of a chimaerical assortment of temnospondyl and pareiasaur teeth (Liu et al., 2014). Although a serrated canine from the Jiyuan Formation could represent an actual gorgonopsian (Liu et al., 2014), this identification cannot be confirmed by this element alone (it could just as easily represent a therocephalian, and indeed Liu et al. (2014) noted its similarity to teeth of the early therocephalian Lycosuchus). As such, the presence of gorgonopsians in the Chinese record remains dubious. Ray & Bandyopadhyay (2003) referred several skull fragments (premaxillary and vomerine elements and isolated dentary symphyses) from the Kundaram Formation of India to Gorgonopsia. These specimens are likely to represent true gorgonopsians: the steep, robust morphology of these symphyses, with the incisor and canine tooth row elevated high above the postcanine row, is typical of the group. However, although this morphology is typical for gorgonopsians it is not exclusive to them; again, some therocephalian taxa exhibit a very similar morphology (Durand, 1991). Of the extra-African regions yielding Permian tetrapod fossils, only Russia has thus far produced definitive gorgonopsian remains.

The history of gorgonopsian discoveries in Russia dates back to the 1890s, beginning with the massive excavations of V. P. Amalitzky at the Sokolki locality along the North Dvina River (Ochev & Surkov, 2000). Amalitzky collected extensive remains of late Permian gorgonopsians, which were initially, briefly described as a new species in a posthumous publication (Amalitzky, 1922). This first Russian gorgonopsian, Inostrancevia alexandri, remains the best-known taxon from the Northern Hemisphere, and has become one of the most famous Permian animals because of its gigantic size (among gorgonopsians, rivaled only by Rubidgea atrox from southern Africa). The genus name of this animal was spelled “Inostranzevia” in its initial description (Amalitzky, 1922), but the spelling Inostrancevia has since come into universal usage and must be maintained (under Art. 33.3.1 of the International Code of Zoological Nomenclature (ICZN, 1999)). Pravoslavlev (1927) subsequently produced a monographic account of the North Dvina gorgonopsians, recognizing three additional species of Inostrancevia (I. latifrons, I. parva, and I. proclivis) and the new genus Amalitzkia (containing two species, A. wladimiri and A. annae).

Hartmann-Weinberg (1938) described another new species of Russian gorgonopsian, albeit placing it in the South African genus Arctognathus: A. progressus from the Tetyushkii District of Tatarstan. Whereas South African gorgonopsian discoveries continued at an alarming pace during this time (Wyllie, 2003), this was mostly a fallow period for Russian gorgonopsian research, with few new specimens being collected. Vjushkov (1953) revised the North Dvina gorgonopsians, recognizing a new genus (Pravoslavlevia) for the small species I. parva (which Efremov (1940) had previously considered to represent a juvenile of one of the larger Inostrancevia species). Bystrov (1955) revised A. progressus, placing it in a new genus, Sauroctonus. Tatarinov (1974) revised all theriodonts from the Soviet Union in a comprehensive monograph on the group, and named a new species of Inostrancevia (I. uralensis) based on an isolated braincase from the Blumental-3 locality of the Orenburg Region.

Additional Russian gorgonopsian taxa were not recognized until the turn of the century, with the description of the small gorgonopsians Viatkogorgon ivakhnenkoi from the Kotelnich locality of the Kirov Region (Tatarinov, 1999a) and Suchogorgon golubevi from the Ust’e Strel’ny locality of the Vologda Region (Tatarinov, 2000a). Most recently, another isolated braincase (from the Klimovo-1 locality of the Vologda Region) was made the holotype of Leogorgon klimovensis, purportedly the first Russian rubidgeine gorgonopsian (Ivakhnenko, 2003; although see Kammerer (2016) for doubts on this identification).

Since the description of Viatkogorgon, no other gorgonopsians have been described from the Kotelnich locality, despite an explosion in the therocephalian diversity reported from this area (Tatarinov, 1999b, 2000b; Kammerer & Masyutin, 2018; Ivakhnenko, 2011). Here, we provide evidence for a second taxon of Kotelnich gorgonopsian based on a complete skull and partial skeleton in the collections of the Vyatka Paleontological Museum, redescribe the cranium of Viatkogorgon for comparison, and place these taxa in a phylogenetic context.

Nomenclatural acts

The electronic version of this article in portable document format will represent a published work according to the International Commission on Zoological Nomenclature (ICZN), and hence the new names contained in the electronic version are effectively published under that Code from the electronic edition alone. This published work and the nomenclatural acts it contains have been registered in ZooBank, the online registration system for the ICZN. The ZooBank Life Science Identifiers (LSIDs) can be resolved and the associated information viewed through any standard web browser by appending the LSID to the prefix http://zoobank.org/. The LSID for this publication is: urn:lsid:zoobank.org:pub:397E7247-DB64-4B99-B24E-5A2E2DA87B48. The online version of this work is archived and available from the following digital repositories: PeerJ, PubMed Central, and CLOCKSS.

Geological Context

The Kotelnich locality consists of a series of Permian red bed exposures along the banks of the Vyatka River in the Kotelnichskii District of Kirov Region. This locality is one of the most productive Permian tetrapod sites in Russia, and is especially well known for its pareiasaur remains (Efremov & Vjushkov, 1955). Although the paleontological importance of the Kotelnich beds has long been recognized (Hartmann-Weinberg, 1937), the geology and stratigraphy of this locality was poorly understood until work by Coffa (1997a, 1997b, 1998, 1999) as part of his PhD research. He recognized five lithological members making up the Kotelnich red beds: (in descending order) the Sokol’ya Gora Member (brown fine-grained fluvial sandstone), Chizhi Member (fine-grained gray sandstone; originally spelled “Chizhy” but emended by Benton et al. (2012)), Shestakovy Member (brown-gray mudstone), Boroviki Member (orange fine-grained aeolian sandstone), and Vanyushonki Member (red-brown calcareous clay and mudstone). More recent studies have generally upheld Coffa’s stratigraphic scheme. Although there has been some disagreement in precise interpretation of the succession (Golubev, 2000; Tverdokhlebov, 2009), the most recent comprehensive stratigraphic and sedimentological overview of the Kotelnich locality (Benton et al., 2012) followed the five-member scheme and nomenclature of Coffa (1999). Benton et al. (2012) also provided the most up-to-date geological maps and stratigraphic sections for Kotelnich, illustrating the positions in section that have yielded vertebrate fossils (Benton et al., 2012: fig. 7).

The Vanyushonki Member is the oldest rock unit in the Kotelnich succession and the source of most of the fossil tetrapods from this locality. Skeletal remains are abundant in this member and often consist of complete, articulated skeletons. The Vanyushonki Member is composed mainly of pale or moderate brown mudstones (clays/silts with small quantities of fine-grained sand). This horizon also contains an inclusion of gray and bluish-gray mudstone and two horizons of dark red mudstone at the base of the exposure. Benton et al. (2012) argued that mudstones of the Vanyushonki Member were probably deposited from suspension in standing water bodies on floodplains or in shallow ephemeral lakes, but noted that in the absence of primary sedimentary structure, the exact depositional environment could not be determined. This member has yielded the most diverse vertebrate assemblage at Kotelnich. In addition to the extremely abundant pareiasaurs (Deltavjatia rossica), the Vanyushonki Member has also produced “nycteroleter” parareptiles (Emeroleter laevis) and various therapsid taxa (including the basal anomodont Suminia getmanovi, the gorgonopsian V. ivakhnenkoi, and the therocephalians Chlynovia serridentata, Gorynychus masyutinae, Karenites ornamentatus, Perplexisaurus foveatus, Scalopodon tenuisfrons, Scalopodontes kotelnichi, and Viatkosuchus sumini) (Ivakhnenko, 2008; Kammerer & Masyutin, 2018; Tsuji, 2013).

Benton et al. (2012) considered the Kotelnich faunal complex to be latest Guadalupian in age, and equivalent to the Pristerognathus Assemblage Zone (AZ) of the South African Karoo Basin. More recent research on the South African AZs, however, indicates that the boundary between the Tapinocephalus and Pristerognathus AZs is latest Guadalupian (260.26 Ma) and the bulk of the Pristerognathus AZ may actually be early Lopingian (Day et al., 2015). Kurkin (2011) suggested that the Kotelnich assemblage was better correlated with the South African Tropidostoma AZ, based on the shared presence of toothed oudenodontid dicynodonts (Australobarbarus in Russia and Tropidostoma in South Africa). However, Australobarbarus fossils are found at a different site (Port Kotelnich) than most of the known Kotelnich vertebrates. As discussed by Benton et al. (2012), the Port Kotelnich outcrop is not easily correlated with the other Kotelnich sections. They suggested, however, that the Port Kotelnich strata are substantially higher in section (probably representing the Shestakovy Member) than the Vanyushonki Member beds that produce the other articulated tetrapod specimens. Based on our experience at Port Kotelnich, we agree that the Australobarbarus-bearing beds are most likely in the Shestakovy Member, and as such are not necessarily indicative of the age of the main Vanyushonki Member fauna.

The specimen KPM 310 (holotype of the new taxon Nochnitsa geminidens described below) was collected by A. Khlyupin in 1994, in a red mudstone of the Vanyushonki Member along the north bank of the Vyatka River. It was collected at the Sokol’ya Gora–Chizhi site, 43 m upstream from the third ravine of the Sokol’ya Gora lens. Following collection, the specimen was mechanically prepared at the Vyatka Paleontological Museum by O. Masyutina.

Systematic Paleontology

Synapsida Osborn, 1903

Therapsida Broom, 1905

Gorgonopsia Seeley, 1894

Nochnitsa gen. nov.

LSID: urn:lsid:zoobank.org:act:8FF18791-BAAD-45AC-946A-722A3BF83139

Type species: Nochnitsa geminidens sp. nov.

Etymology: Named after a nocturnal spirit in Slavic legend (also the namesake for Myotis bats in modern Russian), often portrayed as a horrific female apparition that attacks sleeping humans. Name intended as a regionally-appropriate counterpart to the usual gorgonopsian generic stem “gorgon,” referring to monstrous hags from Greek myth. Feminine.

Diagnosis: As for type and only species.

Nochnitsa geminidens sp. nov.

(Fig. 1–6)

LSID: urn:lsid:zoobank.org:act:DA63D0AC-4592-4E4A-AB19-6E253B0FE5EE

Holotype: KPM 310, a nearly complete skull and lower jaws with articulated vertebrae, ribs, and partial right forelimb from the Kotelnich locality, Kotel’nichskii District, Kirov Region, Russia.

Etymology: From the Latin geminus (“twin”) and dens (“tooth”), referring to the autapomorphic “twinned” sets of postcanines in this species. A noun in apposition.

Diagnosis: Distinguished from all other known gorgonopsians by the autapomorphic “twinning” of its upper postcanines, with the postcanine tooth row consisting of pairs of teeth separated by short diastemata. Further distinguished from the co-occurring gorgonopsian V. ivakhnenkoi by the higher upper postcanine tooth count (six vs. four in Viatkogorgon), upper incisor tooth row nearly in-line with postcanine tooth row (instead of elevated above it), absence of a maxillary flange around the canines, straight (rather than recurved) postcanine crowns, weak mandibular symphysis without a steep, distinct “chin,” proportionally longer snout, smaller orbit, broader intertemporal region, shorter temporal fenestra, dorsoventrally narrower subtemporal arch, absence of a squamosal flange at the posteroventral corner of the temporal fenestra, dorsoventrally narrower dentary ramus, and relatively posterior position of the reflected lamina (mostly beneath the temporal fenestra rather than the orbit). Distinguished from the South African middle Permian gorgonopsian Eriphostoma microdon by the absence of a labial emargination on the maxilla, longer snout, narrower postorbital bar, and higher upper postcanine tooth count (three in Eriphostoma).

Description: The type and only known specimen of N. geminidens (KPM 310) consists of an almost-complete skull, anterior axial column, and right forelimb (Figs. 1 and 2). The skull is relatively small for a gorgonopsian (82 mm dorsal length). The right side of the skull is obscured by the radius, ulna, and autopodial elements (Fig. 2), but the left side is fully exposed (Fig. 1). The skull is generally well preserved (Figs. 3 and 4C), although some sutural details are obscure and the snout has suffered some surface cracking (Fig. 3). Additionally, the left side of the skull is largely undistorted, although the right side has suffered from some lateral compression (Figs. 5 and 6). A large crack extends from the anterior edge of the left orbit across the interorbital region to the right temporal fenestra and has been filled with plaster (Fig. 5); additional small cracks have been filled with silicone rubber (Figs. 3 and 6).

Figure 1 Holotype block of Nochnitsa geminidens (KPM 310).

(A) left lateral view with (B) interpretive drawing. Abbreviations: at, atlas; ax, axis; cr, cranium; cv, cervical vertebra; d, dentary; dv, dorsal vertebra; rla, reflected lamina of angular; r, rib; sc, scapula. Gray indicates matrix. Scale bar equals 1 cm. Photograph and drawing by Christian F. Kammerer.

Figure 2 Holotype block of Nochnitsa geminidens (KPM 310).

(A) right lateral view with (B) interpretive drawing. Abbreviations: ce, centrale; d, dentary; dc, distal carpal; dpc, deltopectoral crest; h, humerus; mc, metacarpal; ph, phalanx; ra, radius; re, radiale; ul, ulna; ue, ulnare. Gray indicates matrix, hatching indicates plaster. Scale bar equals 1 cm. Photograph and drawing by Christian F. Kammerer.

Figure 3 Stereopair of KPM 310, holotype of Nochnitsa geminidens, in left lateral view.

Scale bar equals 1 cm. Photographs by Christian F. Kammerer.

Figure 4 Kotelnich gorgonopsians compared in lateral view.

(A) PIN 2212/6, holotype of Viatkogorgon ivakhnenkoi, in right lateral view with (B) interpretive drawing. (C) KPM 310, holotype of Nochnitsa geminidens, in left lateral view (mirrored for comparative purposes and with non-cranial parts of block edited out) with (D) interpretive drawing. Abbreviations: an, angular; ar, articular; C, upper canine; c, lower canine; d, dentary; fr, frontal; I, upper incisor; i, lower postcanine; j, jugal; la, lacrimal; mf, maxillary flange; mx, maxilla; na, nasal; pa, parietal; PC, upper postcanine; pc, lower postcanine; pf, pineal foramen; pmx, premaxilla; po, postorbital; pof, postfrontal; pp, preparietal; prf, prefrontal; rla, reflected lamina of angular; sa, surangular; sc, sclerotic ring; sf, squamosal flange; smx, septomaxilla; sq, squamosal; ss, squamosal sulcus. Gray indicates matrix, hatching indicates plaster. Scale bars equal 1 cm. Photographs and drawings by Christian F. Kammerer.

Figure 5 Right marginal dentition of Nochnitsa geminidens.

(A) close-up of holotype, KPM 310, with (B) interpretive drawing. Abbreviations: C, upper canine; I, upper incisor; PC, upper postcanine; pc, lower postcanine; rC, replacement upper canine. Scale bar equals 1 cm. Photograph and drawing by Christian F. Kammerer.

Figure 6 Holotype skull of Nochnitsa geminidens (KPM 310) in dorsal view.

(A) interpretive drawing and (B) photograph. Abbreviations: fr, frontal; j, jugal; la, lacrimal; mx, maxilla; na, nasal; nc, nuchal crest; pa, parietal; pf, pineal foramen; po, postorbital; pof, postfrontal; pp, preparietal; prf, prefrontal; smx, septomaxilla; sq, squamosal. Gray indicates matrix, hatching indicates plaster. Scale bar equals 1 cm. Photograph and drawing by Christian F. Kammerer.

The premaxilla is damaged anteriorly, with the internarial bar broken off, and the palatal surface is completely obscured by the occluded mandible (Fig. 4D). The left premaxilla is largely worn off, although the roots of the incisors remain in place (Fig. 2). The right facial portion of the premaxilla has some surficial cracking but is otherwise intact. The premaxilla has a very short contribution to the side of the snout (6 mm out of a 46 mm long snout), being mostly overlapped by the maxilla laterally (Fig. 4D). The premaxillary-maxillary suture is immediately ventral to the septomaxilla, and terminates between I2 and I3 at the alveolar margin of the snout. A total of five upper incisors are present, as in most gorgonopsians. The incisors are mesiodistally narrow and needle-like anteriorly but become progressively apicobasally shorter and mesiodistally broader posteriorly. The upper incisors are weakly recurved and finely serrated on their distal margins (no mesial serrations are evident, although as the mesial margin of these teeth is slightly angled inwards they may be obscured).

The septomaxilla has a broad plate making up the floor of the external naris ventrally, a constricted middle section separating the external naris from the large maxillo-septomaxillary foramen, and a narrow, attenuate facial process extending between the maxilla and nasal (Fig. 4D). Because of damage to the premaxilla at the snout tip, more of the septomaxilla is exposed laterally than would have been present naturally, giving this element a more primitive, “pelycosaurian” appearance than would have been present in the intact skull (compare Fig. 4D with the undamaged septomaxilla of Viatkogorgon in Fig. 4B). Because of damage to the anterodorsal margin of the maxilla, the maxillary-septomaxillary foramen is also larger than would normally have been the case.

The maxilla in Nochnitsa is relatively long and low (Fig. 4D) compared to other gorgonopsians, even similarly long-snouted forms such as Cyonosaurus (Olson, 1937; Sigogneau-Russell, 1989). The lateral surface of the maxilla bears distinct dermal sculpturing in the form of radiating ridges extending outwards from the region around the canine root (Fig. 3). The lateral surface of the maxilla is also densely foraminated, particularly above the tooth row. The posterior process of the maxilla is a narrow, attenuate structure underlying the jugal, which terminates below the midpoint of the orbit. The alveolar margin of the maxilla is weakly convex, with a marked embayment anteriorly at the diastema between the incisors and canine. The canine is relatively small and narrow (18 mm apicobasal length, 6 mm width at base) for a gorgonopsian. The canine is clearly serrated posteriorly, but there is no evidence of anterior serrations (although as for the incisors, because of the angulation of the canines and some matrix covering, the absence of anterior serrations should not be taken as certain). Both functional canines are in the anterior alveoli in this specimen, with replacement canines erupting in the posterior alveoli at the time of death (Figs. 4D and 5). The right replacement canine was more fully erupted—although it is badly damaged, its tip would have reached near mid-height of the right canine (Fig. 4D). The left replacement canine is present only as a newly-erupted tip, shorter than any of the postcanines (Fig. 4D).

A total of six right and five left upper postcanines are present. The morphology of the right upper postcanine tooth row is unique among therapsids, consisting of three pairs of postcanines separated by short (∼2 mm) diastemata (Fig. 5). Furthermore, in each of these pairs the anterior tooth is smaller than its posterior counterpart (lengths of PC1: 5 mm vs. PC2: 7 mm; PC3: 5 mm vs. PC4: 8 mm; PC5: 5 mm, PC6 damaged so length uncertain, but anteroposteriorly broader than PC5). Other than in size, all postcanines are morphologically similar, being elongate, weakly posteriorly-canted but not recurved, and bearing fine posterior serrations (and in these teeth, the anterior face is well-exposed enough to confidently state that anterior serrations are absent). Diastemata breaking up sections of the postcanine tooth row are otherwise unknown in gorgonopsians, but this does not seem to be an artifact of replacement in this specimen, as the gaps between tooth pairs show no signs of empty alveoli or developing crowns. On the left side, the posterior two postcanines (PC5 and 6) are paired in exactly the same way as the teeth on the right side (PC5: 3.5 mm vs. PC6: 6 mm length). The anterior three postcanines do not form similarly distinct pairs, but this seems to be the result of PC1 missing as part of the replacement history of the dentition. If this is the case, then the first preserved left postcanine represents PC2. This tooth is shorter than any other upper postcanine and appears to be newly-erupted—if the pairs erupt in tandem a shorter PC1 may not yet have erupted. Under this interpretation, PC3 and 4 can also be interpreted as a pair: although their bases are not in direct contact as in the other pairs, PC3 (4 mm length) is shorter than PC4 (6 mm length), comparable to the right postcanines.

The nasal is an elongate bone making up the dorsal roof of the snout (Fig. 5A). It bears low anteroposterior ridges but is overall weakly ornamented compared to the maxilla. Its posterior suture with the frontal is situated anterior to the orbits.

The prefrontal is a dorsoventrally low but anteroposteriorly extensive bone at the anterodorsal margin of the orbit (Figs. 4D and 6A). It has a sharp margin at the edge of the orbit but not an expanded or rugose circumorbital rim. Below the prefrontal, the lacrimal is a small, rhomboidal bone. Both of these elements bear irregular, ridged dermal sculpturing. The lacrimal foramen does not exit laterally on the lacrimal, it is restricted to the internal orbital surface.

The jugal has a broad facial portion below the lacrimal (Fig. 4D), but this facial portion is relatively small compared to other gorgonopsians (see, e.g., Viatkogorgon; Fig. 4B). Posterior to this the jugal is a narrow, rod-like bone forming the zygoma. It is overlapped by the postorbital under the postorbital bar. It makes a small contribution to the posterior base of the lateral surface of the postorbital bar and medial face of the bar but does not have an extensive dorsal process participating in the bar as in therocephalians (Durand, 1991; van den Heever, 1994). The jugal is distinctly bowed in the zygoma, so that there is a ventral concavity below the postorbital bar. Posteriorly, the jugal makes up part of the subtemporal bar but is mostly overlain by the zygomatic ramus of the squamosal.

The squamosal has a small contribution to the medial margin of the temporal fenestra dorsally, but is mostly confined to the occiput and zygomatic arch (Fig. 4D). The sutures of the occiput are largely indeterminable in this specimen, but given the proportions of the occiput it is likely that the squamosals made up much of the lateral edge of the occiput as in other gorgonopsians (Fig. 6A). The occipital and zygomatic portions of the squamosal are separated by a sharp ridge with a flange-like ventral edge, anterior to which is a triangular depression, the squamosal sulcus. Restriction of the sulcus to the zygomatic portion of the squamosal is an unusual feature, usually this sulcus extends uninterrupted from the occiput around to the zygoma or is entirely restricted to the occiput (as in rubidgein rubidgeines; Kammerer, 2016). The only other taxon with this morphology is Viatkogorgon (Fig. 4B). The zygomatic ramus of the squamosal terminates in a pointed anterior process dividing the jugal in lateral view; it does not reach the level of the postorbital bar.

The frontal is a hexagonal bone of the interorbital region (Figs. 4D and 6A). It has a broad contribution to the orbit compared to many gorgonopsians (in most rubidgeines it is excluded entirely; Kammerer, 2016). Posteriorly, it terminates in a narrow, sliver-like process extending between the postfrontal and a complementary anterior process of the parietal (Fig. 6A).

The postfrontal is relatively large in Nochnitsa (Fig. 6A), which is typical for basal therapsids but independently modified in most therapsid clades (lost in eutherocephalians, cynodonts, and some anomodonts, and reduced in size in some later gorgonopsians (e.g., Arctognathus); Hopson & Barghusen, 1986; Kammerer, 2015). The postfrontal makes up the posterodorsal corner of the orbit and extends posteriorly between the frontal and postorbital until the level of the anterior border of the pineal foramen.

The postorbital is composed of a laminar dorsal ramus making up the medial border of the temporal fenestra and a rod-like ventral ramus making up almost the entirety of the postorbital bar (Fig. 4D). A weak fossa serving as an attachment site for jaw musculature is present on the ventrolateral edge of the dorsal ramus and continues onto the posterior edge of the postorbital bar. The postorbital bar is weakly curved anteriorly and terminates in only a slight anteroposterior expansion ventrally, unlike the massively expanded ventral postorbital tips of most gorgonopsians (Sigogneau, 1970; Laurin, 1998; Kammerer et al., 2015; Kammerer, 2016).

The preparietal is a roughly rhomboidal median element situated between the frontals and parietals (Fig. 6A). It is flush with the skull roof and unornamented. It is separated from the pineal foramen by a short midline suture of the parietals; it does not abut the foramen directly as in the majority of anomodonts and biarmosuchians (King, 1988; Sidor & Rubidge, 2006).

The parietal is the primary skull roofing bone of the intertemporal region (Fig. 6A). It has attenuate anterior and posterior processes; the latter mirrors the postorbital in following the edge of the temporal fenestra. The anterior portion of the parietal midline is split by the small (3 mm diameter), subcircular pineal foramen. It is surrounded by a distinct, collar-like rim but is not elevated on a “chimney”-like boss as in many basal therapsids.

As mentioned above, the bones of the occiput are not readily differentiable in this specimen. There is a long, well-developed nuchal crest (Fig. 6A) that runs uninterrupted from the top of the occipital plate to the foramen magnum along the midlines of what are probably the postparietal and supraoccipital (based on the configuration seen in other gorgonopsians).

The dentary of Nochnitsa is very unusual for a gorgonopsian, much more closely resembling that of a typical therocephalian. Notably, it lacks a steep, robust symphysis with a distinct mentum, instead having a long, gradually sloping anterior face (Fig. 4D). The dentary ramus posterior to the symphysis is relatively low and narrow, and the coronoid region is only weakly sloped upward, with a convex posterior edge. In most gorgonopsians (including Viatkogorgon; Fig. 4B) the dentary coronoid process is strongly dorsally directed with a distinctly concave posterior face. The roots of four incisors (the standard number for gorgonopsians) can be seen on the damaged right side of the dentary symphysis. They are similar in morphology to their upper counterparts, except apicobasally shorter (crown height 4 mm in i1, vs. 6 mm in I1). The lower canines are mostly obscured by the upper jaw but their bases are visible—these teeth are similar in size to the upper canine and situated anterior and medial to them. The lower postcanines are very similar in individual morphology to the uppers but are not paired; instead, they form a continuous row of close-packed, posteriorly-canted teeth (Fig. 5). A total of six lower postcanines are exposed on each side, but in different parts of the tooth row (the posteriormost left lower postcanines are exposed in the diastema between PC4 and 5, whereas the posteriormost right lower postcanines are exposed in the diastema between PC2 and 3). This suggests that the actual lower postcanine count exceeds six, especially considering that the close spacing of these teeth necessitates more of them for the lower tooth row to approximate the length of the upper.

The only exposed postdentary element is the reflected lamina of the angular, which is best preserved on the left side (Fig. 4D). The entire reflected lamina is intact; it is angled posteroventrally. This lamina is remarkably elongate and narrow (10 mm maximum length), and tapers somewhat ventrally. A single robust ridge runs along the long axis of the reflected lamina. This ridge is common to all gorgonopsians, but usually there is a second, horizontal ridge making a cruciate pattern (Sigogneau-Russell, 1989), which is absent in Nochnitsa. Although the articular is not exposed in this specimen, the proximity of the reflected lamina to the ventral part of the squamosal that borders the quadrate necessitates that this taxon would have the reflected lamina very close to the jaw articulation, which is the primitive condition for therapsids, but unlike most gorgonopsians in which the reflected lamina is separated from the articular by a length of non-laminar angular (Kammerer, 2016).

Part of the postcranium is preserved in articulation with the skull, including the cervicals, some dorsals with associated ribs, and right forelimb (Figs. 1 and 2). The cervical series is curled around the left of the rear of the skull and still partially embedded in matrix (Fig. 1B). The axial spine is broadly rounded and similar in morphology to that of other gorgonopsians (Sigogneau-Russell, 1989; Gebauer, 2014). The dorsals are preserved as fragments of centrum and transverse processes interspersed with ribs. The ribs are simple, elongate elements. Above these ribs on the left side of the specimen, the top of the right scapula is exposed. It is elongate, narrow, and weakly curved, comparable to that of other small gorgonopsians (e.g., Cyonosaurus) but unlike the anteroposteriorly expanded scapular spines of Inostrancevia (Sigogneau-Russell, 1989).

The right humerus, radius, ulna, and most of the manual elements are preserved in partial articulation (Fig. 2B). The humerus is relatively gracile, with a short, weakly-developed deltopectoral crest. The radius and ulna show distinct distal curvature, and the distal tip of the radius forms a discrete edge differentiated from the shaft. No olecranon process is visible on the ulna, but it is possible that this is the result of damage (the proximal tip of this element is not complete and has partially been replaced by mudstone). Preserved proximal carpal elements consist of the radiale, ulnare, and two irregular smaller elements that probably represent the centralia. The ulnare is the proximodistally longest carpal and is expanded at its proximal and distal tips. The radiale is a shorter, more rounded element. The possible centralia are poorly-preserved but appear weakly curved. The concave surface of the centrale would presumably have articulated with the radiale in life, based on the condition in other gorgonopsians (Sigogneau-Russell, 1989). A clear intermedium is not visible; this element is usually small in gorgonopsians and may either be missing or still buried in matrix. Several small, irregular bones between the proximal carpals and metacarpals probably represent distal carpals, but these elements are too poorly preserved to identify further. Based on their great length relative to the other manual elements, the two best-preserved metacarpals probably represent 3 and 4, which are the longest metacarpals in all other gorgonopsians for which manus are known (Sigogneau-Russell, 1989; Kümmell & Frey, 2014). A shorter but still elongate element preserved underneath the ulnare and a possible distal carpal may represent metacarpal 5. A semi-articulated set of poorly-preserved bones appressed to the surface of the humerus and dentary appear to represent fingers, one potentially terminating in the ungual. Based on the size of the phalanx-like elements, these probably correspond to the third and fourth digits, disarticulated from metacarpal 3 and 4. These elements are too poor for a definitive phalangeal count, and there is no clear evidence of the disc-like reduced phalanges typically present in gorgonopsians (Hopson, 1995).

Viatkogorgon Tatarinov, 1999a

Type species: Viatkogorgon ivakhnenkoi Tatarinov, 1999a.

Diagnosis: As for type and only species.

Viatkogorgon ivakhnenkoi Tatarinov, 1999a

(Fig. 4)

Holotype: PIN 2212/61, a complete skeleton from the Kotelnich locality, Kotel’nichskii District, Kirov Region, Russia.

Diagnosis: Distinguished from all other known gorgonopsians by the extremely large squamosal sulcus, extending onto a squamosal flange impinging on the ventral edge of the temporal fenestra. Also characterized by unusually large orbit with proportionally large sclerotic ring. Distinguished from all gorgonopsians other than Nochnitsa by the narrow ventral terminus of the postorbital bar. Further distinguished from Nochnitsa by the suite of features listed in the Diagnosis for that taxon.

Description: The type specimen of V. ivakhnenkoi is one of the most complete gorgonopsian specimens in the world, with nearly the entire postcranium preserved intact, including elements very rarely preserved in therapsid specimens, such as the gastralia. Regrettably, the skull is poor by comparison—although the right side is reasonably well-preserved, the left side and palate are badly broken up and other than the snout tip and left postorbital the skull roof is entirely reconstructed in plaster. Furthermore, the skull as a whole has suffered from lateral compression, making it narrower in dorsal view than it would have been in life. Detailed description of the postcranium of Viatkogorgon will greatly improve our understanding of gorgonopsian skeletal anatomy; unfortunately it was not available for study during the course of the current research (as it was part of a traveling exhibition) and must be dealt with in a future contribution (a preliminary description was provided by Tatarinov (2004)). The following description will focus on comparisons with Nochnitsa, necessarily centering on the overlapping preserved portions of the skull (primarily the lateral surface and mandible).

Like that of Nochnitsa, the premaxilla of Viatkogorgon has limited exposure on the facial surface of the snout (Fig. 4B). Although damaged even on the right side, its suture with the maxilla is visible, and is in a similar position to that of Nochnitsa (below the base of the septomaxillary footplate). The internarial bar is preserved, and is curved slightly posteriorly in lateral view, so that the anterolateral margin of the snout in Viatkogorgon is blunt rather than pointed. The dorsal tip of the premaxilla extends to the anterodorsal edge of the external naris. Few upper incisors are preserved intact, but the typical gorgonopsian count of five appear to be present based on the partial roots and alveoli. They are weakly recurved and spatulate with clear distal serrations. It is uncertain whether they decrease in size posteriorly, as in Nochnitsa, because the only incisors with intact crowns are both interpreted as the I5 of their respective sides.

The septomaxilla of Viatkogorgon has a shorter posterior facial process than that of Nochnitsa (Fig. 4B). Although the facial process separates the nasal and maxilla, as in most therapsids (Hopson & Barghusen, 1986), the absolute tip of the septomaxilla actually extends to overlie the anterodorsal margin of the maxilla. Unlike that of Nochnitsa, the intranarial portion of the septomaxilla is well preserved in Viatkogorgon. The septomaxilla is strongly constricted between the small septomaxillary-maxillary foramen and the main narial opening. The septomaxillary footplate is dorsoventrally shallow and is underlain by a posterior process of the premaxilla. The maxilla is proportionally taller and anteroposteriorly shorter than that of Nochnitsa. Whereas the dorsal margin of the maxilla is broadly rounded in Nochnitsa (Fig. 4D), in Viatkogorgon there is a broad posterodorsal process that extends between the nasal and lacrimal. The posterior process of the maxilla is shorter in Viatkogorgon than Nochnitsa, not reaching the midpoint of the orbit. The lateral surface of the maxilla is damaged on both sides of the skull, but at least some radiating surface ridges were clearly present, as in Nochnitsa (Figs. 4A and 4C). The precanine “step” between the incisors and canine is notably steeper in Viatkogorgon than Nochnitsa, and the canine-bearing portion of the maxilla in general is strongly convex, giving the appearance of a flange in lateral view. The canine is relatively small for a gorgonopsian, similar to Nochnitsa. It is serrated posteriorly. The postcanine tooth row is short, consisting of only four close-packed postcanines. These postcanines are recurved, unlike those of Nochnitsa. The maxilla is weakly emarginated above the postcanine tooth row, to a greater extent than Nochnitsa but not to the degree in Eriphostoma (Kammerer et al., 2015) or rubidgeines (Kammerer, 2016).

The nasal is somewhat broader anteriorly (at the level of the posterior edge of the external naris) in Viatkogorgon than Nochnitsa, although this has been exaggerated in lateral view by lateral compression in the skull (Fig. 4B). The prefrontal of Viatkogorgon is proportionally shorter than that of Nochnitsa and contributes less to the anterodorsal margin of the orbit (which instead has a greater contribution from the lacrimal). The prefrontal has irregular, ragged edges with weak interdigitation with the maxilla anteriorly and lacrimal ventrally. The posterior border of the maxilla is smoothly sloping in Nochnitsa, with progressively shorter contributions to the snout from the prefrontal, lacrimal, and jugal (Fig. 4D). In Viatkogorgon, the lacrimal has an anterior process breaking up the posterior border of the maxilla posteriorly and extending to the same extent as the anterior tip of the prefrontal (Fig. 4B). As mentioned above, the lacrimal extends further dorsally along the rim of the orbit in Viatkogorgon than Nochnitsa. As in Nochnitsa, there is no exit on the lateral surface of the lacrimal for the lacrimal foramen, which is restricted to the orbital wall.

A well-preserved, nearly undistorted sclerotic ring is preserved within the right orbit of Viatkogorgon, consisting of 15 ossicles (Fig. 4B). This ring is remarkably large even within the proportionally very large orbit (outer diameter 2.3 cm, inner diameter 1.5 cm, orbit diameter 2.8 cm), falling well within the range of what is considered scotopic in therapsids (Angielczyk & Schmitz, 2014) and suggesting nocturnal habits for Viatkogorgon. Fragments of the left sclerotic ring are also preserved in the left orbit.

The jugal of Viatkogorgon has a more extensive facial contribution than that of Nochnitsa (Fig. 4B). Its proportions in the zygoma, including the short contribution to the posterior base of the postorbital bar, are very similar to those of Nochnitsa. However, it is substantially taller in the subtemporal bar (which is taller in general than that of Nochnitsa) and more obscured by the more anteriorly-extending squamosal (which extends almost to the level of the postorbital bar.) In lateral view, the ventral portion of the jugal contribution to the subtemporal bar is barely exposed, unlike the condition in Nochnitsa, where it is longer than the dorsal portion (Fig. 4D).

Little of the frontal is preserved in Viatkogorgon; only the portion contributing to the orbital wall is intact (Fig. 4B). Within the orbit, the anterior border of the frontal is bifurcated by a posterior process of the prefrontal. The postfrontal is represented solely by a thin strip of bone at the posterodorsal edge of the orbit; it is otherwise reconstructed in plaster.

The dorsal ramus of the postorbital, like the rest of the intertemporal skull roof, is missing in this specimen (Fig. 4B). The ventral ramus is a thin rod making up the postorbital bar (similar to that of Nochnitsa, but somewhat anteroposteriorly broader). The postorbital bar is a straight rod in lateral view, lacking the anteroventral curvature of Nochnitsa. It also has a weak posterior fossa, presumably for attachment of the jaw muscles.

The zygomatic portion of the squamosal shows the same distinctly bounded squamosal sulcus with posteroventral flange as in Nochnitsa, but is much larger and more expansive, and extends onto an anterodorsal flange at the ventral edge of the temporal fenestra (Fig. 4B). It is also taller anterior to the sulcus and extends further forward on the subtemporal bar. Below the squamosal sulcus the lateral edge of the quadrate is exposed, having disarticulated slightly from the slot it fits into. The quadrate is preserved in articulation with the articular.

Unlike Nochnitsa, Viatkogorgon has the typical gorgonopsian jaw morphology, with a tall mandibular symphysis bearing a distinct mentum (Fig. 4B). The lower dentition is poorly preserved: a single intact incisor and postcanine are exposed on the left side, and the base of the canine is visible on the right, anterior to the upper. The morphology of the lower teeth is very similar to the uppers: the one preserved incisor is recurved and spatulate, the lower postcanine is also weakly recurved, and both have at least posterior serrations. The dentary of Viatkogorgon is generally taller in the ramus than that of Nochnitsa and has a more sharply sloping coronoid process, with the weakly concave posterior edge typical of gorgonopsians. Unfortunately the postdentary bones of Viatkogorgon are badly damaged, such that the morphology of the reflected lamina is almost completely unknown. A narrow strip of this lamina is preserved immediately posterior to the posteroventral edge of the dentary, but it shows no morphology of note. What is evident is that the reflected lamina was situated well anterior to the jaw articulation, like all gorgonopsians other than Nochnitsa. The surangular is exposed as a narrow strip of bone at the top of the jaw posterior to the coronoid process of the dentary. It extends to the articular posteroventrally but the contact between these bones is indistinct. The articular is typical of gorgonopsians, with a ventrally-protruding retroarticular process. Damage to this process makes it uncertain whether it had a hook-like anterior tip as in later gorgonopsians (Kemp, 1969).

Phylogenetic Analysis

Nochnitsa geminidens and Viatkogorgon ivakhnenkoi were included in an expanded version of the most recent published phylogenetic analysis of gorgonopsians, that of Kammerer (2017; itself an expansion of Kammerer (2016)). The original analysis of Kammerer (2016) was focused on rubidgeine gorgonopsians in particular; in order to better understand the relationships of the Kotelnich gorgonopsians, taxon sampling in the current analysis has been expanded to include the Russian taxa Inostrancevia, Pravoslavlevia, Sauroctonus, and Suchogorgon. The early therocephalian Lycosuchus vanderrieti was also included as a representative of the probable sister-taxon of Gorgonopsia, Eutheriodontia (Therocephalia + Cynodontia) (Hopson & Barghusen, 1986).

In addition to expanding the taxon sample, the character matrix has been emended by adding new characters and making alterations to several previous characters. These changes are detailed below:

Character 2: Posterior margin of palatal premaxillary body. Previously (Kammerer, 2016), the states for this character were formulated as (0) gently rounded and (1) with deep invaginations. The biarmosuchian outgroups (Biarmosuchus and Hipposaurus) and the middle Permian gorgonopsian Eriphostoma were the only taxa coded as (0) for this character; all other gorgonopsians were coded as (1), or (?) if the anterior palate was not preserved or exposed. This character was originally intended to encompass an important difference in palatal morphology between gorgonopsians and non-gorgonopsian therapsids. In non-gorgonopsian basal therapsids, such as biarmosuchians, the anterior margin of the choana is gently rounded, with an even, gradual curvature between the body of the premaxilla and the vomerine process of the premaxilla (Fig. 7A). In gorgonopsians, the anterior portion of the choana generally extends further forwards in the form of a narrow channel, producing a distinct invagination between the body of the premaxilla and the vomerine process (see, e.g., Kammerer, 2017: fig. 10). In rubidgeines, this invagination is particularly prominent (Fig. 7B), because the body of the premaxilla is anteroposteriorly expanded relative to the condition in biarmosuchians and early gorgonopsians like Eriphostoma.

Figure 7 Premaxillary-vomerine complex in biarmosuchians and gorgonopsians.

(A) NHMUK R5700, holotype of the biarmosuchian Lycaenodon longiceps. (B) RC 35, holotype of the rubidgeine gorgonopsian Leontocephalus cadlei (considered synonymous with Aelurognathus tigriceps by Kammerer (2016)). (C) PIN 4548/138, a referred specimen of the Russian gorgonopsian Suchogorgon golubevi. All specimens in ventral view, anterior is right. Abbreviations: pi, premaxillary invagination; pmx, premaxilla; v, vomer. Scale bars equal 1 cm. Photographs by Christian F. Kammerer.

Kammerer (2016) coded E. microdon as (0) for this character based on a CT-reconstruction of the palate in the holotype (Kammerer, 2014), although the premaxillary morphology in that specimen is not exactly concordant with that of biarmosuchians. In the Russian gorgonopsians in which this region is exposed (Inostrancevia, Sauroctonus, and Suchogorgon), the premaxillary morphology is very similar to that of Eriphostoma. However, the material of Suchogorgon is much better preserved than that of Eriphostoma and reveals that, although it is not as distinctive as in rubidgeines, an invagination is present at the anterior edge of the choana in that taxon (Ivakhnenko, 2005). One of the “exploded” skulls of Suchogorgon (PIN 4548/138) is particularly informative on this point, as the premaxillary-vomerine complex of this specimen has been isolated and completely prepared (Fig. 7C). This specimen shows the invagination to be morphologically distinct from (but probably representing the ancestral state of) that of rubidgeines. Rather than being an elongate channel between expanded lobes of the premaxillary body and vomerine process, the invagination in Suchogorgon is simply the result of the vomerine process sloping dorsally, leaving a dorsoventral opening between its anterolateral edge and the more ventrally-situated body of the premaxilla. In poorly preserved specimens, such as the holotype of E. microdon, this gap is difficult to see, but re-examination of the CT-scan files for that specimen indicate that is was indeed present. Accordingly, character state (1) has been changed from “with deep invaginations” to simply “invaginated,” with Eriphostoma now coded (1). The distinction between the morphology in Eriphostoma/Suchogorgon and that of rubidgeines appears to be of phylogenetic importance and is worthy of further study. At present it is difficult to encapsulate this distinction in character form, however, because it seems to have undergone gradual transformation in gorgonopsian evolution—in Gorgonops, for instance, the invagination is intermediate in morphology between that of Suchogorgon and that of rubidgeines (see Kammerer, 2015: fig. 12C).

Character 9: Vomerine-pterygoid contact. Previously (as character 7 of Kammerer (2016)), this character had two states, (0) present and (1) absent. The absence of a contact between the vomer and pterygoid is one of the classic gorgonopsian synapomorphies (Hopson & Barghusen, 1986). However, as discussed in further detail below, Russian gorgonopsians with well-prepared palates (Inostrancevia, Sauroctonus, and Suchogorgon) show that they still retained a narrow but clearly present contact between the vomer and pterygoid (Fig. 8). The minimal contact between these bones in these taxa still clearly distinguishes them from non-gorgonopsian therapsids, so this character has been reformatted as ordered multistate: (0) present, broad, (1) present, narrow, and (2) absent.

Figure 8 Photographs and interpretive drawings of the palates of Russian gorgonopsians.

These illustrate the presence of a narrow contact between the vomer and pterygoid. (A), (C) PIN 156/5, holotype of Sauroctonus progressus. (B), (D) PIN 4548/1, referred specimen of Suchogorgon golubevi. Abbreviations: ec, ectopterygoid; mx, maxilla; PCa, postcanine alveolus; pl, palatine; ppl, palatal boss of palatine; ppt, palatal boss of pterygoid; pt, pterygoid; tpt, transverse process of pterygoid; v, vomer. Scale bars equal 1 cm. Photographs and drawings by Christian F. Kammerer.

Character 10: Palatine foramina: (0) small or absent; (1) large, well-developed near maxillary border. New character. In the Russian gorgonopsians Inostrancevia, Sauroctonus, and Suchogorgon, there is a series of large, well-developed foramina on the palatine, near the border with the maxilla (this region is not well preserved in Pravoslavlevia). These foramina appear to be absent in most African gorgonopsian taxa (this portion of the palatine is not exposed in Nochnitsa or Viatkogorgon). The palatine surface is poorly-preserved (or more usually, poorly-prepared) in many African gorgonopsian specimens, so this absence may be partially artifactual. However, some African gorgonopsian specimens with excellently-prepared palates (e.g., UMZC T891, Ruhuhucerberus haughtoni, and SAM-PK-K11458, Arctognathus curvimola) clearly lack large palatine foramina. Comparable foramina are present and well-developed in UMZC T878, however, the specimen of Sycosaurus nowaki that Kemp (1969) described as Leontocephalus intactus.

Character 11: Dentition on palatine boss. Previously (as character 8 of Kammerer (2016)), this character had three states, (0) extensive, (1) single long tooth row, and (2) few teeth in one position. Eutheriodonts such as Lycosuchus lack palatine dentition, necessitating the addition of a fourth state, (3) absent. This character remains ordered, as absence of the palatine dentition represents the endpoint of progressive reduction of these teeth.

Character 17: Parabasisphenoid blade position: (0) restricted to posterior edge of basicranial girder; (1) extending throughout length of basicranial girder. New character. In most gorgonopsians, the parabasisphenoid bears a tall, narrow, blade-like ventral crest. In rubidgeines this “blade” is absent, and the ventral surface of the parabasisphenoid exhibits a reversal to the biarmosuchian condition, in which a narrow median channel separates the two edges of the parabasisphenoid. The presence/absence of this feature is covered in character 15, but does not address an important difference in morphology between the Russian gorgonopsians Inostrancevia, Sauroctonus, and Suchogorgon (the parabasisphenoid rostrum is not well preserved in Pravoslavlevia) and non-rubidgeine African gorgonopsians. In the Russian taxa, the “blade” is nearly semi-circular in lateral view and restricted to the posterior portion of the parabasisphenoid, near the back of the basicranial girder (as is also the case in therocephalians), whereas in the African taxa the “blade” is semi-oval and more elongate, extending for almost the entire length of the basicranial girder (Fig. 9).

Figure 9 Photographs of the basicranial girder in Russian and African gorgonopsians.

These illustrate the difference in morphology of the parabasisphenoid blade between these groups: elongate and sloping in African gorgonopsians (A, B), short and tab-like in Russian gorgonopsians (C, D). (A) BP/1/7275, referred specimen of Eriphostoma microdon. (B) BP/1/4089, referred specimen of Gorgonops torvus. (C) Cast of PIN 2005/1587, holotype of Inostrancevia alexandri. (D) PIN 156/6, holotype of Sauroctonus progressus. Parabasisphenoid blades highlighted in white to show outlines (dotted outline indicates broken surface). (A–C) in ventrolateral view, (D) in lateral view. Photographs by Christian F. Kammerer.

Character 28: Postorbital bar. Previously (as character 24 of Kammerer (2016)), this was treated as an ordered multistate character with three states: (0) unexpanded, (1) expanded (>10% of basal skull length), and (2) greatly expanded (>20% of basal skull length). The primary distinction in this formulation was between non-rubidgeine and rubidgeine gorgonopsians, with the latter having anteroposteriorly expanded (and greatly expanded in the case of rubidgeins) postorbital bars. However, this formulation did not address a difference between gorgonopsians and outgroups like Biarmosuchia. In almost all known gorgonopsians, the ventral tip of the postorbital bar is expanded where it contacts the jugal, even if it is narrow for the rest of its length. The only known exceptions are Nochnitsa and Viatkogorgon, in which the postorbital bar is nearly the same width throughout its length (as in biarmosuchians). To reflect this, the states for this character have been changed to (0) unexpanded (including biarmosuchians, Nochnitsa, and Viatkogorgon), (1) expanded ventrally (non-rubidgeine gorgonopsians), (2) expanded throughout length (>10% of skull length) (non-rubidgein rubidgeines), and (3) greatly expanded throughout length (>20% of skull length) (rubidgeins). The character is retained as ordered, as it reflects increasing levels of expansion of this bone.

Character 29: Facial portion of jugal. Previously (as character 25 of Kammerer (2016)), this character had two states: (0) confluent with suborbital zygomatic portion and (1) depressed relative to zygomatic portion. State 1 was intended to describe the condition in some rubidgeines, where there is a sharp break in surface height between the zygomatic and facial portions of the jugal, with the latter depressed relative to the former. Pravoslavlevia and Inostrancevia also exhibit a depressed facial portion of the jugal, but in a fundamentally different way than in rubidgeines. In rubidgeines, the facial portion of the jugal is depressed relative to the zygoma, but not adjacent facial bones (i.e., the jugal is not strongly depressed relative to the maxilla and lacrimal, the surfaces of these bones are roughly confluent). In Pravoslavlevia and Inostrancevia, there is a broad, deep preorbital fossa composed of the depressed facial surfaces of the lacrimal and jugal. These bones are depressed relative to the adjacent prefrontal and maxilla as well as the zygomatic portion of the jugal. Here, these different styles of facial jugal depression are treated as separate, unordered character states, with the new states for this character being (0) lateral surface confluent with other facial bones and suborbital zygomatic portion of jugal, (1) depressed relative to zygomatic portion but not other facial bones, and (2) bears broad fossa extending onto lacrimal surface, facial portion of jugal depressed relative to both zygomatic portion and other (non-lacrimal) facial bones.

Character 50: Lateral surface of reflected lamina. Previously (as character 45 of Kammerer (2016)), this character had two states: (0) lobate sculpturing and (1) well-developed dorsoventrally-oriented bar, with weakly-developed crossbar. The cruciate laminar sculpturing of state (1) is characteristic of gorgonopsians. Uniquely among gorgonopsians, the reflected lamina of Nochnitsa bears only the main, dorsoventrally-oriented bar; no crossbar is present. As such this character has been changed to ordered multistate to reflect the intermediate condition in Nochnitsa: (0) lobate sculpturing, (1) well-developed dorsoventrally-oriented bar only, and (2) well-developed dorsoventrally-oriented bar, with weakly-developed crossbar.

This expanded version of the Kammerer (2016) data matrix (available as Supplemental Information) is made up of 23 operational taxonomic units (OTUs; all species-level taxa except for Inostrancevia, which was coded at the genus-level due to uncertain alpha taxonomy) and 52 characters. All characters are discrete-state, and of these, six are ordered multistate characters (characters 11, 14, 26, 28, 31, 50). The phylogenetic analysis was run in PAUP* (Swofford, 2002) v4.0 (build 159) using branch-and-bound searching. Bootstrap analysis was done using “fast” stepwise-addition on 1,000 replicates.

The phylogenetic analysis recovered six most parsimonious trees (MPTs) of length 113 (Fig. 10), with a consistency index of 0.566, a retention index of 0.799, and a rescaled consistency index of 0.453. Their topologies differ only in the relative positions of A. curvimola, Lycaenops ornatus, and the clade made up of (Arctops willistoni + Smilesaurus ferox). These taxa are always recovered just outside of Rubidgeinae but vary in precise position, with Arctognathus recovered as the sister-taxon of Rubidgeinae in some MPTs and the (Arctops + Smilesaurus) clade in others (Lycaenops is recovered either as sister to (Arctops + Smilesaurus) or outside a clade containing those taxa and Rubidgeinae). Nochnitsa is recovered as the earliest-diverging gorgonopsian taxon, followed by Viatkogorgon then a large clade containing the remaining gorgonopsian OTUs. This clade is broken into two major subclades, one containing only Russian gorgonopsians (Suchogorgon, Sauroctonus, Pravoslavlevia, and Inostrancevia) and the other containing all the African gorgonopsians included in the analysis.

Figure 10 Results of the phylogenetic analysis, showing the consensus of six most parsimonious trees.

Values at nodes represent bootstrap support. Image by Christian F. Kammerer.

Discussion

The position of Nochnitsa as the basalmost known gorgonopsian is supported by a number of plesiomorphic characters, such as the low symphysis, low, sloping, therocephalian-like posterior dentary, reflected lamina close to jaw articulation, no “cross-bar” in reflected lamina surface, and elongate tooth row (not restricted as in Viatkogorgon). Viatkogorgon is found outside of the clade containing all remaining gorgonopsians, based on its lack of a ventral expansion on the postorbital bar.

Remarkably, the remaining Russian gorgonopsian genera (Inostrancevia, Pravoslavlevia, Sauroctonus, and Suchogorgon) were found to form a monophyletic group outside of the clade containing all African gorgonopsians. The monophyly of this group is currently weakly supported, but their position outside of the “African clade” is supported by several notable characters. The canonical synapomorphy of Gorgonopsia is a midline contact between the palatines, excluding the vomer from contact with the pterygoid; previous researchers considered this morphology to be present in all gorgonopsians (Hopson & Barghusen, 1986; Sidor, 2000). However, our examination of the well-preserved holotype skull of Sauroctonus progressus (PIN 156/6) revealed that, contra previous descriptions (Tatarinov, 1974; Sigogneau-Russell, 1989; Gebauer, 2014), there is actually a narrow vomerine-pterygoid contact (Figs. 8A and 8C). Although this contact is substantially narrower than in any other therapsid clade, it clearly prevents midline contact of the palatines. Further examination of the best-preserved Russian gorgonopsian palatal material indicates that this contact is also retained in Suchogorgon (Figs. 8B and 8D) and Inostrancevia (the palate is too poorly preserved in the only known skull of Pravoslavlevia to determine).

African gorgonopsians typically have a conservative morphology of the parabasisphenoid: this element bears a blade-like ridge extending the length of the basicranial girder, from the anterior edge of the basal tubera to the posterior edge of the transverse processes of the pterygoids (Figs. 9A and 9B). This blade is tallest in its posterior half but slopes gradually forwards, the only exception being in rubidgeines in which the blade has been secondarily lost. In Inostrancevia, Sauroctonus, and Suchogorgon, by contrast (this region is not well preserved in Pravoslavlevia), the parabasisphenoid blade is a tab-like structure restricted to the posterior part of the basicranial girder (Figs. 9C and 9D). This is more similar to the condition in therocephalians than to African gorgonopsians (van den Heever, 1994), and may represent the ancestral morphology within Theriodontia.

Based on the results of the phylogenetic analysis, we recognize two geographically-restricted major subclades of Gorgonopsia: a “Russian clade” containing all Russian gorgonopsians other than Nochnitsa and Viatkogorgon, and an “African clade” containing the South African and east African gorgonopsians in the data set (Fig. 10). Although there are numerous African gorgonopsian taxa that have yet to be included in a phylogenetic analysis (pending ongoing alpha taxonomic revision of the group), personal examination of those taxa by the lead author indicates that they all have “African clade”-style midline palatine contacts and elongate parabasisphenoid blades, suggesting that they also belong to this clade. The recovery of a monophyletic group containing the majority of Russian gorgonopsian taxa is novel to the current study. Previous authors recognized no close relationship between Inostrancevia, Sauroctonus, and Suchogorgon (although Inostrancevia and Pravoslavlevia have often been grouped together, and were initially placed in the same genus). Tatarinov (1974) classified the Russian gorgonopsians in different families, with Inostrancevia and Pravoslavlevia in Inostranceviidae (containing only these two genera) and Sauroctonus in subfamily Cynariopinae of family Gorgonopidae (also containing the African genera Aloposaurus, Aloposauroides, Cynarioides, Cynariops, Scylacognathus, Scylacops, and Sycocephalus). Sigogneau-Russell (1989) classified Inostrancevia and Pravoslavlevia in subfamily Inostranceviinae of family Gorgonopidae (one of only two gorgonopsian subfamilies she recognized, the other being Rubidgeinae), leaving Sauroctonus as an undifferentiated gorgonopid. Ivakhnenko (2003) classified Inostrancevia in a monogeneric Inostranceviidae, placing Pravoslavlevia, Sauroctonus, and Suchogorgon in Gorgonopidae. Gebauer (2007) found Inostrancevia to be deeply nested within African gorgonopsians, forming the sister-taxon of Rubidgeinae. She recovered Sauroctonus as a basal gorgonopid, but also deeply-nested within a clade of otherwise-African taxa (as she considered Aloposaurus, Cyonosaurus, and Aelurosaurus to be basal, non-gorgonopid gorgonopsians).

At present, no described gorgonopsian specimens exhibiting the features of the “African clade” have been found in Russia, and vice versa. The purported Russian rubidgeine L. klimovensis is not recognizable as a rubidgeine, and may not even be gorgonopsian (Kammerer, 2016). Gebauer (2014) considered the Tanzanian gorgonopsian originally known as Scymnognathus parringtoni (Huene, 1950) to be referable to the genus Sauroctonus. However, personal examination of the type and only specimen of S. parringtoni (GPIT/RE/7113) indicates that it has the typical parabasisphenoid and palatine morphologies of other African gorgonopsians, and that cranial similarities between it and Sauroctonus (which consist primarily of proportional characters that are variable among gorgonopsians) are superficial. Endemism in gorgonopsians was previously unsuspected, and is surprising considering the many Russo-African sister-taxon relationships in other therapsid groups, particularly dicynodonts and burnetiamorphs (Sidor & Smith, 2007; Kammerer, Angielczyk & Fröbisch, 2011). However, tetrapod biogeography in the Permian remains poorly understood, and the inferred dispersal abilities of various therapsid taxa are often discordant with the observed record (Sidor et al., 2013; Kammerer, Bandyopadhyay & Ray, 2016). Additional research, particularly from regions outside of the well-sampled Karoo Basin of South Africa, is required to understand the factors underlying tetrapod distribution during this time.

Conclusion

Two distinct gorgonopsian taxa are now known from the Russian Kotelnich locality: V. ivakhnenkoi Tatarinov, 1999a and N. geminidens gen. et sp. nov. Despite this addition to the fauna’s gorgonopsian diversity, gorgonopsians remain notably less species-rich at Kotelnich than therocephalians (Ivakhnenko, 2011). Low diversity and small size of the Kotelnich gorgonopsians suggests a predatory therapsid assemblage comparable to that of the Pristerognathus AZ in the Karoo (Kammerer et al., 2015), prior to the main burst of gorgonopsian diversification in South Africa. The phylogenetic position of Nochnitsa and Viatkogorgon is also intriguing in this regard, as they are recovered as the most basal gorgonopsians in our analysis. It is unlikely that this is actually indicative of an earlier age for the Kotelnich locality than middle Permian gorgonopsian-bearing strata in South Africa (e.g., the Abrahamskraal Formation, which yields specimens of Eriphostoma), based on the other therapsid components of these faunas (notably the abundant eutherocephalians and absence of dinocephalians at Kotelnich). However, even though the main “Russian clade” of gorgonopsians probably had diverged by the time of the Kotelnich fauna, its absence at the locality does suggest it had not yet undergone substantial diversification in Russia, only later becoming the dominant therapsid predators in the region.

Supplemental Information

Supplemental Information 1 Character matrix for phylogenetic analysis.

Click here for additional data file.

We are very grateful to director A. Toporov, curator T. Berestova, and the staff members of the Vyatka Paleontological Museum for permitting access to the holotype of Nochnitsa and for supporting research at Kotelnich. For access to comparative materials we thank Valeriy Golubev and the late Mikhail Ivakhnenko (PIN), Sifelani Jirah and Bernhard Zipfel (BP), Ingmar Werneburg (GPIT), Paul Barrett (NHMUK), Robert and Marion Rubidge (RC), Zaituna Erasmus and Roger Smith (SAM), and Matt Lowe (UMZC). Finally, thanks to Adam Huttenlocker, Arjan Mann, and Christian Sidor for reviewing the manuscript for this paper.

Institutional abbreviations

BP Evolutionary Studies Institute, University of the Witwatersrand, Johannesburg, South Africa

GPIT Paläontologische Sammlung, Eberhard-Karls-Universität Tübingen, Germany

KPM Vyatka Paleontological Museum, Kirov, Russia

NHMUK The Natural History Museum, London, UK

PIN Paleontological Institute of the Russian Academy of Sciences, Moscow, Russia

RC Rubidge Collection, Wellwood, Graaff-Reinet, South Africa

SAM Iziko: The South African Museum, Cape Town, South Africa

UMZC University Museum of Zoology, Cambridge, UK

Additional Information and Declarations

Competing Interests

Author Contributions

Data Availability

New Species Registration

The authors declare that they have no competing interests.

Christian F. Kammerer conceived and designed the experiments, performed the experiments, analyzed the data, contributed reagents/materials/analysis tools, prepared figures and/or tables, authored or reviewed drafts of the paper, approved the final draft.

Vladimir Masyutin contributed reagents/materials/analysis tools, authored or reviewed drafts of the paper.

The following information was supplied regarding data availability:

The character matrix for the phylogenetic analysis is provided as a Supplemental File.

The following information was supplied regarding the registration of a newly described species:

Publication LSID: urn:lsid:zoobank.org:pub:397E7247-DB64-4B99-B24E-5A2E2DA87B48;

Nochnitsa: urn:lsid:zoobank.org:act:8FF18791-BAAD-45AC-946A-722A3BF83139;

Nochnitsa geminidens: urn:lsid:zoobank.org:act:DA63D0AC-4592-4E4A-AB19-6E253B0FE5EE

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
