# Peer review of "Gorgonopsian therapsids (Nochnitsa gen. nov. and Viatkogorgon) from the Permian Kotelnich locality of Russia"

_PeerJ, doi:10.7717/peerj.4954_

## Round 0.1 · original submission · Minor Revisions

Dear authors,

I am sorry for the delay in this decision. I have accepted the decision of 'minor revisions' from the three reviewers. However, all three reviewers have made comments about the figures. As such, these will need to be revised before your manuscript can be accepted for publication.

I have some additional comments that the authors should address prior to resubmission (in addition to those made by the reviewers):

1. Authority and date should be provided for each species-level taxon at first mention. Please ensure that the nominal authority is also included in the reference list.
2. Please provide the descriptive statistics for the phylogenetic analysis (CI, RI, RC).
3. Please replace 'tree' with 'cladogram' or 'topology' where appropriate.

Once again, thank you for submitting your manuscript to PeerJ and I look forward to receiving your revised submission.

·

Basic reporting

This is a very solid paper that makes several important contributions, including the recognition of an endemic Russian clade of gorgonopsians and anatomical details on the palate of the Russian species. Below I raise four substantive issues to be addressed in a revision, plus a bunch of minor edits.

Comment 1: Figures 4 and 5 should be combined, so the reader doesn't need to go back-and-forth between the two. In other words, the all of comparative figures should include both photos and drawings on one page (as already done for Figures 8 and 9).

Comment 2: Gorgonopsians and therocephalians are typically considered sister taxa, and the authors seem to agree with this view, given the comparisons they make (e.g., Lines 651-652). However, the outgroup used for the phylogenetic analysis is Biarmosuchia (viz. Hipposaurus and Biarmosuchus). No rationale is provided.
The phylogenetic analysis should be re-run with a mid-Permian therocephalian used as the first outgroup (e.g., Lycosuchus or Glanosuchus). If nothing changes in the results, a line can be added to this effect in the revised manuscript. However, if the results change substantially, then more substantial edits will be needed.

Comment 3: Based on the results of the phylogenetic analysis, the authors propose to recognize two major subclades with Gorgonopsia: Inostrancevioidea and Gorgonopioidea. First, this the phylogenetic results need to be confirmed when the appropriate outgroup is used (see Comment 2). Second, the authors fail to propose a phylogenetic definition for the new taxa, suggesting the need to wait for more robust support from subsequent analyses, before doing so. You can't have it both ways-either provide phylogenetic definitions or don't propose the clade names. Using a new clade name but not providing a definition is a disservice to future workers. If the authors want to stick with their cautious approach, using "Russian clade" and "African clade" as place holders in the discussion is totally appropriate.

Comment 4: Seeing as this paper names a new species, describing the holotype in detail is very important. Unfortunately, the current description of the postcrania is very superficial and should be supplemented in the revision.

The remainder of my comments are minor line edits or suggestions:

L33-36: Smiley et al. (2008) identified a gorgonopsian from the Permian of Niger. It might be worth mentioning this occurrence, since it falls nicely between the southern and northern records mentioned.

Smiley, T. M., C. A. Sidor, O. Ide, and A. Maga. 2008. Vertebrate fauna of the Upper Permian of Niger. VI. First evidence of a gorgonopsian therapsid. Journal of Vertebrate Paleontology 28:543-547.

L55-56: Rephrase as: The first Russian species, Inostrancevia alexandri, remains the best-known gorgonopsian from the… (clearly it is not the best known taxon)

L66-67: Explain or rephrase "at an alarming pace"

L212-215: What is shown in Figure 5 for the septomaxilla is unlike the condition observed in most therapsids. This should be better explained in the text. Is the area damaged? Is it real and a transitional morphology from what is seen in pelycosaur-grade synaspids to what is seen in therapsids?

L214: Add "(i.e., facial") between posterior and process. Facial process is the standard term.

L238: Replace "identical" with "similar" (They can't be a different size and identical)

L235-248: Please discuss any evidence available that the 'twinning' of postcanine teeth on the right side is real. For example, does preparation shown that no alveoli are present in between the pairs of teeth? I'm a little concerned that one of the diagnostic features of the species is apparently absent on the left side of the specimen. More explanation or justification seems appropriate.

L297: Replace "mid-parietal suture" with "midline suture between the parietals"

L307: Replace "from comparison with" with "based on the configuration seen in"

L318: Add "and medial" after "anterior"

L368: Be more specific than "noted above"

L396-397: The description about the septomaxilla and maxilla is unclear (e.g., maxilla is described as "taller and shorter") and should be rewritten. What is the anterodorsal portions of the maxilla?

L472: How can there be 2 citations for the "only existing" analysis? Rephrase to "the only published cladistic analysis of gorgonopsians (Kammerer 2017)."

L474: Insert "to" before "better"

L488-489: Rephrase for clarity. I think you're referring to the body of the premaxilla.

L490: Sidor (2003) not in literature cited.
Sidor, C. A. 2003. The naris and palate of Lycaenodon longiceps (Therapsida: Biarmosuchia), with comments on their early evolution in the Therapsida. Journal of Paleontology 77(5):153-160.

L490: Delete "internal" It's either choana or internal naris.

L513-518: I was a little confused by this section. A new figure showing the 3 character states would definitely benefit future workers.

L541-543: This is confusing as written. Rephrase to: "in Sycosaurus nowacki (UCMZ T878), a specimen Kemp (1969) previously described as Leontocephalus intactus."

L578-579: This is a little confusing because "height" isn't being used in standard vertical orientation here. Please rewrite.

L626, L702: Rephrase to avoid using ambiguous terms like "one node up" and "main burst".

L653-660: See comment #3.

L707: Rephrase to "specimens of Eriphostoma" to avoid using a taxon name as an adjective.

Experimental design

No comment.

Validity of the findings

See comment about choice of outgrip in the phylogenetic analysis. Otherwise, no comment.

Additional comments

Interesting paper and good job.

·

Basic reporting

I have commented on my opinions on the figures, I wish them to be altered from their current state but in general they are still relatively good, and do not dramatically dampen the manuscript.

Experimental design

no comment

Validity of the findings

no comment

Additional comments

Peer Review:
Gorgonopsian therapsids (Nochnitsa gen. nov. and
Viatkogorgon) from the Permian Kotelnich locality of Russia

The authors Kammerer and Masyutan, present extremely important new findings of a basal gorgonopsians from a relatively underworked locality the Kotelnich of Russia. They anoint the new species Nochnitsa geminidens and provide a long awaited re-description of the taxon Viatkogorgon ivakhnenkoi (Tatarinov, 1999).

There are some minor corrections/suggestions I placed in the manuscript, however there are a few major points to bring up here. First, what of the postcrania of Viatkogorgon, how does this compare to Nochnitsa geminidens, I would have like to have seen this described too, it seems gorgon postcranial remains are all to glanced over.

Additionally, and I find most important that there are two major subgroups distinguished here (or recognized) these being the Inostrancevioidea (containing the Russian taxa Inostrancevia, Pravoslavlevia, Sauroctonus, and Suchogorgon) and Gorgonopioidea (containing all African gorgonopsians). While these groups are likely valid, that is I do generally agree with the author, there is an issue in recognizing these clades and not formally diagnosing them here, or even attempting to do as much. I understand that definitions may change, but currently the Rubidgeinae (a large clade of robust African gorgons) also lacks a formal definition, and this whole business of naming clades without providing diagnoses does not sit well with me, It would even be better to just refer to them as an un-named clade until this can be done formally. I suggest a tentative diagnosis is made, including all uniting characters and phylogenetic definitions at present, at least this gives the reader a sense of real support for the naming of these groups.

I feel it would be more than helpful to have a locality map and strat column given the emphasis here of the locality. To be honest, I don’t know why the authors really didn’t do this already since they added a geologic context section.

This leads me to my final point, in general the figures could have been made with more care, particularly some of the line drawing are not very helpful to interpret the anatomy, unless some attempt is made to show depth, with stippling I would say that these should be revised.

If these changes are made I would gladly recommend this article for publication.

Sincerely,

Arjan Mann

·

Basic reporting

In this paper, the authors describe two basal gorgonopsians from the important Kotelnich locality in Russia’s Permian, one of them providing the basis of a new genus and species. The paper further examines the complicated evolutionary relationships of gorgons with emphasis on their global radiation during the middle-to-late Permian transition.

The finding that Russian Inostranceviinae diverged early is an interesting concept, and the overall updated taxonomy and new cladistic definitions will be an important contribution. I do, however, suggest more clarity in the definitions, and I address this in the section below (‘Experimental Design’).

In sum, the project is well researched, the paper is structured well and the existing illustrations are composed well, although I would like to see better use of illustrations (including a geology figure and more labeling of the cladogram Fig. 10). Overall, I would recommend this study for publication with only some minor tweaks.

Experimental design

Geology and Age:
My limited understanding of the Kotelnich locality is that there are fossil-bearing deposits at multiple levels that are either middle or early-late Permian (or both). Fossils are distributed vertically throughout the lower member, and the new gorgon KPM 310 is said to come from the middle of the lower member. It would be most useful if this could be illustrated in a stratigraphic log, but I understand if the data are unavailable for this level of detail. The authors should consider figuring the geologic/stratigraphic context if possible.

Choices on Taxonomy:
Unfortunately, the definitions aren’t as explicit as they could be, further muddying how paleontologists discuss these groups. For example, the text refers to “Gorgonopids” and “Inostranceviids/Inostranceviines,” but these nodes aren’t labeled on the cladogram in Figure 10. Is “Inostranceviidae” equivalent to the node labeled “Inostrancevioidea” in Figure 10?

Two options: (1) For the sake of preserving the term “Gorgonopidae” as it frequently appears in the literature, I would recommend defining the family as inclusively as possible, encompassing the common ancestor of Inostranceviinae and Rubidgeinae and all of its descendants (specifically, the node labeled ‘66’ in Fig. 10). That would only exclude Nochnitsa and Viatkogorgon, essentially reassigning them as non-gorgonopid gorgonopsians. Another option (2) would be to constrain the family by the genus Gorgonops (e.g., all more closely related to Gorgonops than to Eriphostoma and Inostranceviids), but then that would taint the previous literature that has called all these other taxa gorgonopids as well. Hence, my pitch to go inclusive.

Whichever option the authors choose, the important thing is that the definitions are clear and that the cladogram in Figure 10 is adequately labeled. Then we can all start speaking the same language.

Data Repository:
Because gorgon cladistic analyses are so few, I could imagine this matrix being useful to many paleo researchers. The authors should find a repository for the Nexus file, include a complete character list (not just a discussion of state changes from previous publications, which makes replication tedious and difficult), and include a statement or link in the manuscript. MorphoBank would be ideal.

Validity of the findings

I generally agree with the findings, but I think more effort in demonstrating the age of these occurrences (aided by a figure) would give greater confidence in the timing of the early gorgonopsian radiation (is it middle Permian? Is it early-late Permian?). See my comment above on the geology of Kotelnich and its major members.

Additional comments

Minor comments are copied below and marked-up in the attached PDF:

Intro:
lines 29-31 – Gorgonopsians are not “abundant” (see Smith et al., 2012, Forerunners of Mammals: Chapter 2) nor are they “omnipresent.” Rephrase this.

Lines 660 – Clarify: “Belong to [the African gorgonopioid] clade”

Lines 679-694 – The relevant biogeographic discussions of therocephalians by Huttenlocker et al. (2015, JVP) Huttenlocker & Sidor (2016, JVP) and Huttenlocker & Smith (2018, PeerJ) are missing from the ‘Discussion’ section. For example, the 2015 and 2016 papers are invoked in the mention of sister-taxon relationships, but are not cited. How does the gorgonopsian pattern mirror (or not mirror) those already discussed in therocephalians? I would emphasize that it is different in the respect that the gorgons show fewer sister-taxon pairs, but the early dichotomy of major groups is congruent with the finding that so many therocephalian subgroups also include basal Russian representatives. I don’t know what this means, but it’s something to chew on.

Additional References:

Smith, R., Rubidge, B., & Van der Walt, M. (2012). Therapsid biodiversity patterns and paleoenvironments of the Karoo Basin, South Africa. Forerunners of Mammals: Radiation, Histology, Biology. Indiana University Press, Indianapolis, Indiana, 30-62.

Huttenlocker, A. K., Sidor, C. A., & Angielczyk, K. D. (2015). A new eutherocephalian (Therapsida, Therocephalia) from the upper Permian Madumabisa Mudstone Formation (Luangwa Basin) of Zambia. Journal of Vertebrate Paleontology, 35(5), e969400.

Huttenlocker, A. K., & Sidor, C. A. (2016). The first karenitid (Therapsida, Therocephalia) from the upper Permian of Gondwana and the biogeography of Permo-Triassic therocephalians. Journal of Vertebrate Paleontology, 36(4), e1111897.

Huttenlocker, A. K., & Smith, R. M. (2017). New whaitsioids (Therapsida: Therocephalia) from the Teekloof Formation of South Africa and therocephalian diversity during the end-Guadalupian extinction. PeerJ, 5, e3868.

---

## Round 0.2 · accepted · Accept

Dear author,

I have accepted the reviewers decision of ‘accept’.

Once again thank you for choosing PeerJ as your publishing venue, and I hope you use us again soon.

# ·

Basic reporting

I have read the response to reviewers and the revised manuscript and feel that the authors have done an adequate job revising the paper. I have no additional concerns and am happy to see the paper accepted for publication.

Experimental design

No comment.

Validity of the findings

No comment.

Additional comments

Good job. Looks ready for publication.

·

Basic reporting

The writing is clear, and unambiguous.

Experimental design

The research questions are well defined and relevant, the taxonomy is sufficiently adressed.

Validity of the findings

The data is well presented and taxonomic conclusions are fine.

Additional comments

After reading through the script again, the edits the authors have made to all three reviewers seem acceptable to me.

I am glad the phylogenetic comments have been rectified which appear to have been unanimous among the reviewers.

I would note some points on the reviewers interpretation of my comments on figure 9. Perhaps there are no better specimens in the authors mind, but a few specimens of Lycaenops also show this well and conspicuously. My comment here is driven by the really not all that differing morphology between B-C, D could be real but looks broken. If the author is right about this character, that is fine. I just wonder how variable this is within species or through ontogeny. It is not worth holding the paper up on that note however. I defer to the authors here.

Also, I still disagree with your use of Hag, I think some classicists might take offence too (especially relying on the Websters-). If you have studied the field (of Classics (also not necessarily Myth), then you might know the severity of language and interpretations in their discipline (ie the subject of entire Phd's). Daemon, is still the most correct term- I ran this by some classicists at U of T. But thats again a personal opinion on my part and will not hold up this contribution.

thanks for the opportunity to review the study.

-Arjan Mann